# Design, Synthesis, Docking Study, and Antiproliferative Evaluation of Novel Schiff Base–Benzimidazole Hybrids with VEGFR-2 Inhibitory Activity

**DOI:** 10.3390/molecules28020481

**Published:** 2023-01-04

**Authors:** Hany M. Abd El-Lateef, Mohammed A. I. Elbastawesy, Tamer Mohamed Abdelghani Ibrahim, Mai M. Khalaf, Mohamed Gouda, Mariam G. F. Wahba, Islam Zaki, Martha M. Morcoss

**Affiliations:** 1Department of Chemistry, College of Science, King Faisal University, Al-Ahsa 31982, Saudi Arabia; 2Department of Chemistry, Faculty of Science, Sohag University, Sohag 82524, Egypt; 3Department of Pharmaceutical Organic Chemistry, Faculty of Pharmacy, Al-Azhar University, Assiut 71524, Egypt; 4Community Engagement Development Administration, Vice-Presidency for Studies, Development and Community Service, King Faisal University, Al-Ahsa 31982, Saudi Arabia; 5Faculty of Social Work, Helwan University, Helwan 11795, Egypt; 6Department of Pharmacology and Toxicology, Faculty of pharmacy, Nahda University, Beni-Suef 62513, Egypt; 7Pharmaceutical Organic Chemistry Department, Faculty of Pharmacy, Port Said University, Port Said 42526, Egypt; 8Department of Pharmaceutical Chemistry, Faculty of Pharmacy, Nahda University, Beni-Suef 62513, Egypt

**Keywords:** benzimidazole, Schiff, hybrids, docking, VEGFR, anti-proliferative, apoptosis

## Abstract

A new series of Schiff–benzimidazole hybrids **3a**–**o** has been designed and synthesized. The structure of the target compounds was proved by different spectroscopic and elemental analysis tools. The target compounds were evaluated for their in vitro cytotoxic activity against 60 cancer cell lines according to NCI single- and five-dose protocols. Consequently, four compounds were further examined against the most sensitive lung cancer A549 and NCI-H460 cell lines. Compounds **3e** and **3g** were the most active, achieving 3.58 ± 0.53, 1.71 ± 0.17 and 1.88 ± 0.35, 0.85 ± 0.24 against A549 and NCI-H460 cell lines, respectively. Moreover, they showed remarkable inhibitory activity on the VEGFR-2 TK with 86.23 and 89.89%, respectively, as compared with Sorafenib (88.17%). Moreover, cell cycle analysis of NCI-H460 cells treated with **3e** and **3g** showed cellular cycle arrest at both G1 and S phases (supported by caspases-9 study) with significant pro-apoptotic activity, as indicated by annexin V-FITC staining. The binding interactions of these compounds were confirmed through molecular docking studies; the most active compounds displayed complete overlay with, and a similar binding mode and pose to, Sorafenib, a reference VEGFR-2 inhibitor.

## 1. Introduction

Heterocycles, especially heterocycles containing nitrogen, are considered the most fundamental structural units in medicinal chemistry research and development [1,2]. Among several heterocyclic structures, syntheses and biological applications of benzimidazole have been subjected to various studies from many research groups [3,4,5,6].

On another chemical front, the Schiff hydrazone (C=N-) framework is an exceptionally adaptable drug-like moiety which has recently been utilized in the development of cancer treatments or cellular apoptosis [7,8]. The structures containing this bioactive moiety showed remarkable anticancer activities via the restraint of numerous kinds of enzymes, proteins, and/or receptors that plays essential roles in cell growth and survival [9]. Consequently, Schiff-fused chemical compounds, because of their wide scope of biological activities and synthetic applications, have been developed as a target heterocyclic framework in the research and development of medicinal chemistry. (Compound **1**, **3**, Figure 1) [10,11]

Controlling the angiogenesis process is the basic factor of the growth or inhibition of the majority of cancer cell types [12,13]. Angiogenesis is simply defined as the process by which new blood capillaries have arisen from the existing vasculature, and this process can be stimulated through the activation of various chemical signals [14]. Tyrosine kinases (TKs) are one of the main regulators of tumor angiogenesis [15,16]. It has been reported that many TKs receptors, especially VEGFR-2, are over-expressed in many cancer cells. In response to its stimulation, VEGFR-2 can promote a consecutive series of successive signals that control cell growth, survival, and proliferation [17]. The over-stimulation of VEGFR-2 was reported in plenty of cancer cell types when compared to normal cells [18]. Consequently, and depending on the previous facts, many medicinal chemistry researchers target the inhibition of such an enzyme in order to obtain more safe or more selective candidates that treat or operate on cancer angiogenesis with no effect on normal cells [19]. The process of hindering the VEGF pathway can be carried out by blocking the VEGFR-2 receptors activation using novel VEGFR-2 inhibitors [20].

Apoptosis simply means the regulation of programming cell death; it plays a fundamental role in normal cell development or tissue homeostasis [21]. Many researchers showed that the apoptotic process is closely related to the survival of tumors. Moreover, caspases are broadly known of their role in the apoptotic process [22]. Dysregulation of apoptotic caspases leads directly to the inactivation or over-activation of bioactive substrates, the generation of a cascade of signaling events, authorized the controlled demolition of cellular components, and proliferation; therefore, targeting the cellular apoptotic cascades of cancer cells has gained great attention due to its clinically beneficial effects in cancer therapy [23].

Recently, Meguid et al. reported the design and synthesis of a new series of benzimidazole derivatives with promising dual inhibition action on EGFR and VEGFR-2 kinases [24]. Among their tested compounds, compound 2 (see Figure 1) displayed remarkable inhibition activity against VEGFR-2 with IC_50_ 69.62 μM. Furthermore, a notable cytotoxicity activity was found against the HeLa cancer cell line with IC_50_ of 1.44 μM compared to the standard [24].

Finally, and in continuation of our previous research and efforts to synthesize novel heterocyclic compounds bearing various bioactive scaffolds of efficient anticancer activity [25,26], the present work depends on the hybridization of the well-known biologically active entities Schiff base and benzimidazole scaffolds. Our synthesized compounds were tested as anti-cancer candidates targeting VEGFR-2 inhibitors with the analysis of their apoptotic anti-proliferative activities, in the hope that this synergistic hybridization may lead to a more selective, safe, and efficient drug.

## 2. Results

### 2.1. Chemistry

Target compounds **3a**–**o** were chemically synthesized as shown in Figure 1. Compounds **1a**,**b** were prepared according to the reported methods [27]. The chemical structures of such compounds were proven by matching their spectral data with the reported data [27]. The key intermediates compounds, benzimidazole-hydrazone **2a**,**b**, were prepared by heating compounds **1a**,**b** with hydrazine hydrate at reflux temperature for 12 h. The final target benzimidazole–Shiff hybrids **3a**–**o** were prepared by reaction of the benzimidazole hydrazone derivatives **2a**,**b** with the appropriate acetophenone or aldehyde derivatives in ethanol in the presence of few drops of acetic acid as catalyst under reflux conditions. The spectral data, in addition to elemental analyses, showed that all derivatives of **3a**–**o** underwent the reaction smoothly to give the predicted Schiff base hybrid structure in good yields. The ^1^H NMR spectra experienced the disappearance of NH_2_ signal, which indicates the condensation process. On other hand, the appearance of doublet signals with their characteristic pattern in the aromatic δ6.70–8.00 ppm region augments the formation of new hybrids.

As a representative example, compound **3m** NMR spectra showed a broad singlet signal at δ_H_ 10.39 ppm for phenolic OH proton. Moreover, as mentioned, a characteristic pattern for the additional four phenyl protons at δ_H_ 8.04 and 7.06 ppm represented di para-substituted phenyl. Moreover, 1H singlet signal at δ 4.20 is distinctive as an NH proton and could be exchanged using D_2_O, in addition to the singlet 3 proton of the methoxy group at δ 3.83. The distinguished hydrazone proton appeared at δ 8.64 ppm. Moreover, the ^13^C spectrum showed a characteristic peak appearing at δ_C_ 148.09, representing the (C=N) imine carbon. Moreover, the obtained elemental results of compound **3m** align with the calculated data for such a compound.

### 2.2. Biology

#### 2.2.1. Primary In Vitro One-Dose Anticancer Assay

Benzimidazole compounds **3a**–**o** were selected by the National Cancer Institute (NCI), Bethesda, USA, according to the protocols of drug evaluation for in vitro anticancer screening [16]. Primary in vitro one-dose screening assay was performed in full panels of 60 human tumor cell lines derived from nine tumor cell lines, including leukemia, melanoma, lung, colon, CNS, ovarian, renal, prostate, and breast cancer cell lines. The selected benzimidazole hybrids were added at a single concentration (10^−5^ M) and the culture was incubated for 48 h. Analysis of the single dose 60 cell panel assay results showed that benzimidazole molecules **3a**, **3b**, **3d**, **3f**, **3l** and **3n** were found to be inactive while other benzimidazole molecules were active against some tumor cell lines. Benzimidazole molecules **3h**–**k** and **3m** displayed moderate cell growth inhibition activity only against leukemia CCRF-CEM, HL-60(TB) and RPMI-8226 cell lines with percentage growth between 21.39–39.31%. Compound **3c** achieved high cell growth inhibitory activity against leukemia CCRF-CCEM, HL-60(TB), K-562, MOLT-4 and RPMI-8226, non-small cell lung cancer NCI-H23, NCI-H522, colon cancer HCC-2998, KM-12, SW-620, CNS cancer U-251, melanoma MALME-3M, M14, SK-MEL-5, ovarian cancer OVCAR-8, renal cancer RXF 393, SN 12C and breast cancer BT-549 with percentage growth values between 1.86–11.73%. A complete cell death was recorded for the leukemia SR, non-small cell lung cancer NCI-H460, colon cancer COLO 205, HCT-116, HT-29, melanoma LOX IMVI, and breast cancer MCF-7 and MDA-MB-468 cells where the growth percent was −3.17, −43.84, −50.64, −37.23, −0.83, −52.93, −4.44 and −32.95%, respectively. The obtained data revealed an obvious activity profile for the compound **3e** toward leukemia K-562, SR, non-small cell lung cancer HOP-62, colon cancer HCT-116, CNS cancer SF-268, SF-295, melanoma SK-MEL-28, ovarian cancer OVCAR-4, OVCAR-8, renal cancer 786-0, and breast cancer HS-578T cancer cell lines with growth percentage 4.08–14.80%. Compound **3e** revealed complete cell death against non-small lung cancer HOP-92, NCI-H460, CNS cancer U-251, melanoma LOX IMVI, MALME-3M, and renal cancer RXF 393 cells with growth percentage −2.97, −47.73, −51.51, −26.04, −9.02, −17.67%, respectively. In addition, compound **3g** displayed marvelous anticancer activity against non-small cell lung cancer NCI-H23, CNS cancer SF-539, melanoma SK-MEL-28, ovarian cancer IGROV 1, OVCAR-4, renal cancer ACHN, and breast cancer MCF-7 cancer cells where the growth percentage values were in the range of 0.75–12.48%. Additionally, compound **3g** showed complete cell death toward non-small cell lung cancer A549/ATCC, NCI-H460, colon cancer HCT-116, HT-29, CNS cancer U251, melanoma LOX IMVI, MALME-3M, renal cancer 786-0, RXF 393, and prostate cancer DU-145 cells with growth percentage values of −3.93, −64.39, −21.05, −58.89, −62.08, −12.27, −49.93, −32.04, −12.30 and −22.13%, respectively Furthermore, compound **3o** showed remarkable anticancer activity against leukemia K-562, MOLT-4, non-small cell lung cancer EKVX, NCI-H226, CNS cancer SNB-19, melanoma MDA-MB-435, UACC-257, UACC-62, and breast cancer with growth percentage values between 0.08–11.93%. Furthermore, compound **3o** displayed complete cell death toward most of the remaining cell lines with growth percentage values between −5.56 to −86.78%. The obtained results indicate that compounds **3c**, **3e**, **3g**, and **3o** exhibited the highest ability to inhibit the growth of different cancer cell lines compared to other benzimidazole compounds. It could be concluded that Schiff bases attached to the benzimidazole ring might contribute to the activity of the prepared compounds. The presence of an electron donating group (OH or OCH_3_) or an electron withdrawing group (Cl) has better anticancer activity against cancer cell lines over the unsubstituted phenyl derivatives. In addition, replacement of arylidenehydrazono substituent with the arylethylidene group resulted in sharp decrease in growth percentage inhibition. Furthermore, regarding groups present at C-2 of benzimidazole ring, 4-chlorophenyl exhibited the higher activity among the tested groups, indicating that the common C-2-(4-chlorophenyl) has a better contribution in growth percentage inhibitory activity than the 4-hydroxyphenyl group (see the Appendix A).

#### 2.2.2. Full In Vitro Five-Dose Anticancer Assay

Benzimidazole–Schiff hybrids **3c**, **3e**, **3g**, and **3o**, which demonstrated noticeable activities against most tested cell lines, were additionally selected for the five doses testing against the full panel of 60 human tumor cell lines by NCI. In the context, compound **3c** showed high activities against most of the tested cell lines with GI_50_ values ranging from 1.46–7.97 µM. The compound exhibited GI_50_ values in the range of 1–3 µM in 53 tested subpanels. The highest growth inhibition activity was observed against leukemia HL-60(TB) with a GI_50_ value of 1.46 µM. On the other hand, compound **3e** showed obvious sensitivity toward leukemia cell lines, except for the leukemia SR cell line (GI_50_ value 8.94 µM) with GI_50_ values ranging from 1.05–2.04 µM. Concerning non-small cell lung cancer, the compound showed high activity against A549, HOP-62, NCI-H23, and NCI-H522, with GI_50_ values less than 2 µM. All the remaining subpanels showed marvelous sensitivity profiles with GI_50_ not more than 2.56 µM. In addition, compound **3g** displayed GI_50_ values ranging from 1.62–4.30 µM against all the tested subpanels except for the leukemia cancer CCRF-CEM cell line where it showed a GI_50_ value of 6.91 µM. The highest growth inhibitory activity was observed against the renal cancer RXF-393 cell line with a GI_50_ value of 1.62 µM. The obtained data revealed a good sensitivity profile for benzimidazole molecule **3g** toward colon cancer HT-29 and renal cancer UO-31 subpanels with a GI_50_ value of 1.70 µM. Furthermore, compound **3o** showed high anticancer activity against most of the tested cell lines with GI_50_ values between 1.02–9.95 µM. With regard to sensitivity against some individual subpanels, compound **3o** revealed noticeable activity against NCI-H226, SF-539, SNB-19, and SK-MEL-28, with GI_50_ values 1.23, 1.02, 1.03, and 1.09 µM, respectively. Moreover, the obtained data showed high activity against ovarian cancer OVCAR-5 and breast cancer MDA-MB-231 subpanels with GI_50_ values of 1.40 and 1.93 µM, respectively (see the Appendix A).

#### 2.2.3. MTT Assay against Lung Cancer Cell Lines

To determine IC_50_, benzimidazole hybrids **3c**, **3e**, **3g**, and **3o** were further analyzed using the standard MTT colorimetric assay against lung cancer A549 and NCI-H460 cell lines [28], Appendix B. Sorafenib was chosen as a reference control in the present study. The choice of lung cancer cell lines was based on their sensitivity to tested compounds in the NCI-60 cell line assay. The in vitro results showed that the test benzimidazole molecules showed significant anticancer activity against test lung cancer cell lines (Table 1). Benzimidazole molecules **3e** and **3g** showed 1.3–2.1-fold more potent cytotoxic activity than Sorafenib. The cytotoxic activity correlation of the test benzimidazole molecules showed that compounds **3e** and **3g** showed more potent cytotoxic effects as concluded from their IC_50_ values against test lung cancer cell lines when compared to compounds **3c** or **3o**. It could be concluded that, regarding aryl groups present at C-2 of benzimidazole ring, 4-chlorophenyl moiety was favorable and exhibited higher cytotoxic activity than the 4-hydroxyphenyl group.

#### 2.2.4. In Vitro VEGFR-2 Inhibition Assay

VEGFR-2, a key endothelial RTK, functions as a major positive signal transducer for both physiological and pathological angiogenesis. Regulation of VEGFR-2 activation is one of the major important mechanisms that are essential for proteostasis in endothelial cells under pathological conditions [29]. In order to prove the mechanism of the antiproliferative activity of the prepared benzimidazole molecules, compounds **3e** and **3g** were subjected to in vitro VEGFR-2 inhibition activity using ELISA analysis [30]. Sorafenib is a reference VEGFR-2 inhibitor and was included as positive control in this study. The results showed that compounds **3e** and **3g** displayed significantly decreased VEGFR-2 activity compared with untreated NCI-H460 cells. Benzimidazole hybrids **3e** and **3g** subsequently showed 86.23 and 89.89% VEGFR-2 inhibition activity, which were nearly equipotent to or more potent than reference Sorafenib (88.17% inhibition activity). In conclusion, data indicate that benzimidazoles **3e** and **3g** were potent inhibitors of VEGFR-2 in NCI-H460 cells and 4-chlorobenzylidene diazinyl benzimidazole **3g** exhibited higher VEGFR-2 inhibitory activity than 4-methoxybenzylidene diazinyl benzimidazole **3e**, (Figure 2).

#### 2.2.5. Cell Cycle Analysis of Benzimidazole Hybrids **3e** and **3g**

The effect of the most potent compounds **3e** and **3g** on NCI-H460 lung cell line was studied. Treatment of NCI-H460 cells with benzimidazole molecules **3e** and **3g** at a concentration equal to their IC_50_ concentration dose values (IC_50_ = 1.71 and 0.85, subsequently), resulted in significant alteration in cellular cycle phases [10] (Figure 3). A significant increase in the percentage of cells at G1 phase (61.31 and 64.42, subsequently) compared with untreated control (58.70%) was observed. Furthermore, an increase in the S phase percent (**3e**: 35.63%; **3g**: 33.28%) compared with untreated NCI-H460 cells (29.46%) confirmed that both benzimidazole molecules induce cell growth arrest at both G1 and S phases. Additionally, a concurrent reduction in the percentage of cells at G2/M phase (**3e**: 3.06%; **3g**: 2.30%) compared with untreated lung cells (11.83%) was observed. From the obtained results, it could be concluded that the tested benzimidazole hybrids inhibit the cell proliferation through cellular cycle arrest at both G1 and S phases.

#### 2.2.6. Annexin V/ FITC Apoptosis Staining Assay

Induction of apoptotic cascade in cancerous cells is a crucial determinant in the outcome of therapy. The mode of cellular death induced by compounds **3e** and **3g** was further studied to declare whether death is due to apoptosis or necrosis. Accordingly, compounds **3e** and **3g** were selected to be further investigated for their impact on induction of apoptosis in the NCI-H460 lung cancer cell line. The NCI-H460 cells were treated with compounds **3e** and **3g** for 24 h at a concentration equal to their IC_50_ value and then analyzed for the apoptosis percentage via FACS detection using Annexin V-FITC and PI dual staining [25]. As illustrated in Figure 4, the tested cancer cells displayed an increase in the percentage of apoptotic cells following exposure treatment with compounds **3e** and **3g** as observed with untreated control cells. NCI-H460 cells treated with compounds **3e** and **3g** showed 11.17% and 4.36% of cells in the early apoptotic cells, respectively. Additionally, the percentage of late apoptotic cells in NCI-H460 cells after treatment with compounds **3e** and **3g** was 16.54% and 20.74%, respectively. Furthermore, the percentages of early and late apoptotic cells in the untreated NCI-H460 cells were 0.46% and 0.28%, respectively. All the obtained results indicate that benzimidazole hybrids **3e** and **3g** could induce a marked apoptosis in NCI-H460 lung cancer cells.

#### 2.2.7. Caspase Assay

Apoptosis is a programmed cellular death and is a crucial regulator of physiological growth. The activation of apoptosis signal transduction pathways in cancerous cells is the main mechanism of action of the current available chemotherapy or immunotherapy [31]. To investigate whether the cytotoxic activity of benzimidazole [32,33] compounds **3e** and **3g** against NCI-H460 lung cancer cells is secondary to its ability to activate apoptotic cascade, H-460 cells were treated with compounds **3e** and **3g** at a concentration equal to their IC_50_ concentration for 48 h and then subjected to ELISA analysis as shown in Figure 5. The results showed that compounds **3e** and **3g** caused a significant increase in the level of active *c*aspase-9 compared to untreated H-460 cells. It is worth mentioning that benzimidazole molecules **3e** and **3g** were 13.31 and 11.73-fold more than untreated NCI-H460 cells. The results suggested that the sample induced apoptosis in NCI-H460 cells via activation of *c*aspase-9.

## 3. Docking Study

Based on promising antiproliferative activity shown by test compound **3g** against cancer lines and inhibitory activity exerted on the VEGFR-2 enzyme, we ran molecular docking simulations to explore the possible mode of inhibition of such a class of compounds. Docking simulations within the active site of VEGFR-2 (PDB ID: 4ASD) revealed an interesting binding profile that could be used as an explanation for their inhibitory activity. Table 2 shows that the docking scores of **3g** with the best antiproliferative activity were better than the co-crystallized ligand. Compound **3g** was fitted within the VEGFR-2 active site with high affinity (−8.59 Kcal/mol) in comparison with Sorafenib (−8.74 kcal/mol) (Table 2).

Moreover, Visual inspection of the best docking poses of the test compound revealed a number of binding interactions with key amino acid residues which could help predict its mode of inhibition. Compound **3g** showed good binding through an aryl ring attached to the hydrazone moiety via hydrophobic interaction with Phe 1047 amino acid. In addition to another notable connection via H-donor and H-acceptor, the bioactive hydrazone moiety of **3g** was also bounded to Glu 885 and Asp 1046 through C24 and N23, respectively (Figure 6). Thus, the molecular docking results suggest that compounds **3g** may be bonded to the VEGFR-2 active site with the same manner as reference Sorafenib (Figure 6 and Figure 7).

## 4. Conclusions

In conclusion, a new series of Schiff base–benzimidazole molecules was designed so as to obtain potential anticancer hybrids. The designed compounds were constructed and structurally confirmed on the basis of ^1^H-NMR and ^13^C-NMR spectroscopy as well as elemental microanalysis. All the constructed benzimidazole hybrids were selected for their in vitro anticancer activity according to the NCI-60 single-dose analysis. Compounds **3c, 3e, 3g**, and **3o** were further selected for NCI five-dose anticancer analysis. Results indicated that benzimidazole molecules **3c**, **3e**, **3g**, and **3o** were potent anticancer agents showing broad spectrum anticancer activity against the tested human cancer subpanels. The anticancer activity of tested benzimidazole hybrids is correlated to VEGFR-2 enzyme inhibition where benzimidazoles **3e** and **3g** were potent VEGFR-2 enzyme inhibitors (86.23 and 89.89% inhibition activity, respectively) as compared with Sorafenib (88.17% inhibition activity). Moreover, the cellular cycle flow cytometry analysis demonstrated that benzimidazole molecules **3e** and **3g** induce cellular cycle arrest at both G1 and S phases of NCI-H460 cells. In addition, benzimidazole molecule is a strong inducer of apoptosis as elicited by the results of the Annexin staining analysis. Furthermore, ELISA measurements for *c*aspase-9 showed that benzimidazole hybrids **3e** and **3g** boosted the level of active caspase-9 by 13.31 and 11.73-fold, respectively, compared with untreated NCI-H460 cells. In addition, the binding interaction of **3g** was also confirmed through molecular docking studies; the compound displayed complete overlay with, and a similar binding mode and pose to, Sorafenib, a reference VEGFR-2 inhibitor.

## 5. Experimental Section

### 5.1. Chemistry

General details: refer to (Appendix A). Synthesis and analytical data of compounds **1** and **2** were as reported [27].

#### 5.1.1. General Synthesis of Compounds **3a**–**o**

To a suspension of benzimidazole hydrazone **2** (1.0 mmol) in ethanol (20 mL), an appropriate aryl ketone or aryl aldehyde derivative (1.0 mmol) with 2 drops of glacial acetic acid was added. The reaction mixture was heated under reflux for 8–10 h. After cooling, the formed precipitate was filtered, washed with diethyl ether, and crystallized from ethanol to give pure compound **3a**–**o** (see Appendix A).

##### 6-((*Z*)-((*E*)-Benzylidenehydrazono)(phenyl)methyl)-2-(4-chlorophenyl)-1*H*-benzo[d]imidazole (**3a**)

Yield: 0.34 g (79%); mp: 210–211 °C,^1^H NMR (400 MHz, DMSO-*d*_6_) *δ* ppm 8.72 (s, 1H, HC=N), 8.23 (d, *J =* 8.1 Hz, 2H, Ar-H), 8.10 (s, 1H, Ar-H), 7.88 (d, *J =* 7.4 Hz, 1H, Ar-H), 7.84–7.69 (m, 9H, Ar-H), 7.60 (t, *J* = 7.6 Hz, 2H, Ar-H), 7.53 (d, *J* = 7.8 Hz, 2H, Ar-H), 4.92 (s, 1H, NH). ^13^C NMR (101 MHz, DMSO-*d*_6_) *δ* ppm 163.18, 148.67, 142.48, 141.84, 139.94, 135.60, 134.65, 134.41, 134.34, 133.53, 130.74, 129.37, 128.80, 127.88, 127.67, 12.698, 126.72, 123.83, 119.16, 111.84. Anal. Calcd for C_27_H_19_ClN_4_ (434.92): C, 74.56; H, 4.40; N, 12.88. Found: C, 74.66; H, 4.49; N, 12.75.

##### 2-(4-Chlorophenyl)-5-((*Z*)-phenyl((*E*)-(1-phenylethylidene)hydrazono)methyl)-1*H*-benzo[d]imidazole (**3b**)

Yield: 0.30 g (68%); mp: 200–202 °C,^1^H NMR (400 MHz, DMSO-*d*_6_) *δ* ppm 8.23 (t, *J =* 7.6 Hz, 3H, Ar-H), 8.01 (s, 1H, Ar-H), 7.84–7.72 (m, 10H, Ar-H), 7.61 (d, *J* = 7.4 Hz, 2H, Ar-H), 7.53 (s, 1H, Ar-H), 5.21 (s, 1H, NH), 1.11 (s, 3H, CH_3_). ^13^C NMR (101 MHz, DMSO-*d*_6_) *δ* ppm 162.67, 159.30, 152.07, 149.48, 142.05, 135.55, 134.69, 134.22, 133.24, 129.07, 128.65, 127.82, 126.65, 124.76, 124.12, 121.85, 118.66, 113.13, 112.26, 112.26, 111.13, 21.77. Anal. Calcd for C_28_H_21_ClN_4_ (448.95): C, 74.91; H, 4.71; N, 12.48. Found: C, 75.01; H, 4.50; N, 12.65.

##### 4-((*E*)-((*Z*)-((2-(4-Chlorophenyl)-1*H*-benzo[d]imidazol-6-yl)(phenyl) methylene)hydrazono)methyl)phenol (**3c**)

Yield: 0.37 g (84%); mp: 178–180 °C, ^1^H NMR (400 MHz, DMSO-*d*_6_) *δ* ppm 8.56 (s, 1H, HC=N), 8.24 (d, *J =* 7.8 Hz, 2H, Ar-H), 8.01 (s, 1H, Ar-H), 7.88 (d, *J =* 7.4 Hz, 1H, Ar-H), 7.84–7.69 (m, 10H, Ar-H), 7.60 (t, *J* = 6.8 Hz, 2H, Ar-H), 4.41 (s, 1H, NH). ^13^C NMR (101 MHz, DMSO-*d*_6_) *δ* ppm 164.40, 160.45, 148.24, 148.02, 145.36, 141.42, 135.63, 133.42, 129.70, 129.45, 128.92, 128.44, 128.26, 127.90, 127.67, 124.82, 124.56, 124.29, 124.18, 123.89. Anal. Calcd for C_27_H_19_ClN_4_O (450.92): C, 71.92; H, 4.25; N, 12.43. Found: C, 71.71; H, 4.33; N, 12.45.

##### 4-((*E*)-1-((*Z*)-((2-(4-Chlorophenyl)-1*H*-benzo[d]imidazol-5-yl)(phenyl)methylene)hydrazono)ethyl)phenol (**3d**)

Yield: 0.38 g (81%); mp: 181–183 °C,^1^H NMR (400 MHz, DMSO-*d*_6_) *δ* ppm 8.90 (s, 1H, OH), 7.55 (d, *J* = 8.3 Hz, 2H, Ar-H), 7.47 (d, *J* = 6.9 Hz, 1H, Ar-H), 7.07–7.00 (m, 7H, Ar-H), 6.82 (s, 2H, Ar-H), 6.77–6.70 (m, 3H, Ar-H), 6.57 (d, *J* = 8.6 Hz, 1H, Ar-H), 4.82 (s, 1H, NH), 1.80 (s, 3H, CH_3_). ^13^C NMR (101 MHz, DMSO-*d*_6_) *δ* ppm 162.26, 155.06, 151.72, 151.01, 146.45, 138.58, 131.61, 129.65, 129.56, 128.95, 128.67, 128.27, 127.02, 126.61, 122.06, 116.61, 116.48, 116.28, 114.86, 114.03, 113.49, 21.52. Anal. Calcd for C_28_H_21_ClN_4_O(464.95): C, 72.33; H, 4.55; N, 12.05. Found: C, 72.11; H, 4.37; N, 11.95.

##### 2-(4-Chlorophenyl)-6-((*Z*)-((*E*)-(4-methoxybenzylidene)hydrazono)(phenyl)methyl)-1*H*-benzo[d]imidazole (**3e**)

Yield: 0.36 g (82%); mp: 182–184 °C, ^1^H NMR (400 MHz, DMSO-*d*_6_) *δ* ppm 8.63 (s, 1H, HC=N), 8.23 (d, *J* = 8.4 Hz, 2H, Ar-H), 8.04 (s, 1H, Ar-H), 7.88 (d, *J* = 8.5 Hz, 1H, Ar-H), 7.82–7.78 (m, 6H, Ar-H), 7.71 (t, *J* = 7.4 Hz, 1H, Ar-H), 7.60 (t, *J* = 7.6 Hz, 3H, Ar-H), 7.05 (d, *J* = 8.7 Hz, 2H, Ar-H), 4.79 (s, 1H, NH), 3.83 (s, 3H, OCH_3_). ^13^C NMR (101 MHz, DMSO-*d*_6_) *δ* ppm 162.16 160.94, 152.52, 152.42, 138.21, 138.06, 137.03, 136.80, 132.93, 131.28, 130.47, 130.03, 130.00, 129.52, 129.01, 126.99, 126.42, 125.98, 117.86, 115.20, 114.87, 55.86. Anal. Calcd for C_28_H_21_ClN_4_O (464.95): C, 72.33; H, 4.55; N, 12.05. Found: C, 72.21; H, 4.48; N, 12.14.

##### 2-(4-Chlorophenyl)-5-((*Z*)-((*E*)-(1-(4-methoxyphenyl)ethylidene)hydrazono) (phenyl)methyl)-1*H*-benzo[d]imidazole (**3f**)

Yield: 0.36 g (76%); mp: 198–200 °C,^1^H NMR (400 MHz, DMSO-*d*_6_) *δ* ppm 8.22 (d, *J =* 7.0 Hz, 3H, HC=N and Ar-H), 7.99 (s, 1H, Ar-H), 7.87–7.64 (m, 11H, Ar-H), 7.60 (t, *J* = 7.6 Hz, 3H, Ar-H), 3.68 (br s, 1H, NH), 3.49 (s, 3H, OCH_3_), 2.51 (s, 3H, CH_3_). ^13^C NMR (101 MHz, DMSO-*d*_6_) *δ* ppm 163.14, 161.14, 150.77, 147.22, 146.17, 146.17, 136.53, 135.10, 134.24, 133.64, 133.01, 130.90, 130.51, 129.57, 129.46, 129.28, 128.96, 128.81, 128.58, 126.70, 123.32, 56.52, 21.81. Anal. Calcd for C_29_H_23_ClN_4_O (478.97): C, 72.72; H, 4.84; N, 11.70. Found: C, 72.51; H, 4.58; N, 11.45.

##### 6-((*Z*)-((*E*)-(4-Chlorobenzylidene)hydrazono)(phenyl)methyl)-2-(4-chlorophenyl)-1*H*-benzo[d]imidazole (**3g**)

Yield: 0.38 g (82%); mp: 201–203 °C, ^1^H NMR (400 MHz, DMSO-*d*_6_) *δ* ppm 8.25 (d, *J =* 7.6 Hz, 3H, HC=N and Ar-H), 8.01 (s, 1H, Ar-H), 7.96–7.79 (m, 9H, Ar-H), 7.63 (t, *J* = 7.00 Hz, 1H, Ar-H), 7.61–7.58 (m, 3H, Ar-H), 4.49 (s, 1H, NH). ^13^C NMR (101 MHz, DMSO-*d*_6_) *δ* ppm 164.40, 148.62, 148.24, 148.02, 145.36, 141.21, 136.10, 135.63, 129.70, 129.45, 129.06, 128.92, 128.85, 128.44, 128.26, 129.90, 124.82, 124.56, 124.18, 124.04, 123.89. Anal. Calcd for C_27_H_18_Cl_2_N_4_ (469.36): C, 69.09; H, 3.87; N, 11.94. Found: C, 69.11; H, 3.53; N, 11.75.

##### 4-((*E*)-1-((*Z*)-((2-(4-Chlorophenyl)-1*H*-benzo[d]imidazol-5-yl)(phenyl)methylene)hydrazono)ethyl)aniline (**3h**)

Yield: 0.38 g (81%); mp: 181–183 °C, ^1^H NMR (400 MHz, DMSO-*d*_6_) *δ* ppm 8.21 (d, *J =* 7.4 Hz, 2H, Ar-H), 8.01 (s, 1H, Ar-H), 7.84–7.69 (m, 10H, Ar-H), 7.48–7.46 (m, 3H, Ar-H), 7.41 (s, 2H, NH_2_), 4.43 (s, 1H, NH), 2.52 (s, 3H, CH_3_). ^13^C NMR (101 MHz, DMSO-*d*_6_) *δ* ppm 161.09, 151.88, 146.14, 136.00, 134.08, 133.14, 130.64, 129.64, 128.98, 128.71, 128.53, 128.07, 126.76, 126.12, 121.42, 117.87, 117.25, 115.08, 114.13, 113.77, 111.26, 21.59. Anal. Calcd for C_28_H_22_ClN_5_ (463.96): C, 72.48; H, 4.78; N, 15.09. Found: C, 72.19; H, 4.67; N, 15.25.

##### 4-(6-((*Z*)-((*E*)-benzylidenehydrazono)(phenyl)methyl)-1*H*-benzo[d]imidazol-2-yl)phenol (**3i**)

Yield: 0.34 g (83%); mp: 217–219 °C,^1^H NMR (400 MHz, DMSO-*d*_6_) *δ* ppm 10.56 (s, 1H, OH), 8.09 (s, 1H, HC=N), 8.06 (d, *J =* 7.0 Hz, 2H, Ar-H), 8.00 (s, 1H, Ar-H), 7.83–7.78 (m, 6H, Ar-H), 7.74 (t, *J* = 7.4 Hz, 2H, Ar-H), 7.62–7.59 (m, 3H, Ar-H), 7.07–7.04 (m, 3H, Ar-H), 3.92 (s, 1H, NH). Anal. Calcd for C_27_H_20_N_4_O (416.47): C, 77.87; H, 4.48; N, 13.45. Found: C, 77.66; H, 4.39; N, 13.70.

##### 4-(5-((*Z*)-Phenyl((*E*)-(1-phenylethylidene)hydrazono)methyl)-1*H*-benzo[d]imidazol-2-yl)phenol (**3j**)

Yield: 0.33 g (78%); mp: 201–203 °C,^1^H NMR (400 MHz, DMSO-*d*_6_) *δ* ppm 10.55 (s, 1H, OH), 8.09 (d, *J =* 7.4 Hz, 2H, Ar-H), 8.00 (s, 1H, Ar-H), 7.84–7.77 (m, 6H, Ar-H), 7.70 (t, *J* = 7.2 Hz, 2H, Ar-H), 7.60–7.58 (m, 3H, Ar-H), 7.03 (d, 3H, Ar-H), 4.21 (s, 1H, NH), 2.54 (s, 3H, CH_3_). ^13^C NMR (101 MHz, DMSO-*d*_6_) *δ* ppm 161.86, 153.46, 138.22, 138.02, 137.57, 136.92, 136.62, 132.97, 130.27, 130.02, 129.34, 129.02, 128.70, 128.41, 126.13, 121.05, 117.90, 116.80, 116.60, 115.04, 115.04, 114.60, 21.45. Anal. Calcd for C_28_H_22_N_4_O (430.50): C, 78.12; H, 5.15; N, 13.01. Found: C, 78.01; H, 5.30; N, 12.95.

##### 4-(6-((*Z*)-((*E*)-(4-hydroxybenzylidene)hydrazono)(phenyl)methyl)-1*H*-benzo[d]imidazol-2-yl)phenol (**3k**)

Yield: 0.35 g (82%); mp: 188–190 °C, ^1^H NMR (400 MHz, DMSO-*d*_6_) *δ* ppm ^1^H NMR (400 MHz, DMSO-*d*_6_) δ 10.52 (s, 1H, OH), 8.08 (d, *J* = 8.5 Hz, 3H, HC=N and Ar-H), 7.99 (s, 1H, Ar-H), 7.79 (s, 5H, Ar-H), 7.70 (t, *J* = 7.2 Hz, 2H, Ar-H), 7.60 (d, *J* = 7.5 Hz, 3H, Ar-H), 7.04 (d, *J* = 8.5 Hz, 3H, Ar-H), 4.53 (s, 1H, NH). ^13^C NMR (101 MHz, DMSO-*d*_6_) *δ* ppm 163.42, 157.10, 156.49, 152.92, 15195, 151.38, 148.83, 135.54, 129.68, 129.02, 128.94, 128.87, 122.08, 112.26, 111.54, 111.50, 110.94. Anal. Calcd for C_27_H_20_N_4_O_2_ (432.47): C, 74.98; H, 4.66; N, 12.95. Found: C, 74.81; H, 4.63; N, 12.85.

##### 4-((*E*)-1-((*Z*)-((2-(4-Hydroxyphenyl)-1*H*-benzo[d]imidazol-5-yl)(phenyl)methylene)hydrazono)ethyl)phenol (**3l**)

Yield: 0.32 g (72%); mp: 187–189 °C, ^1^H NMR (400 MHz, DMSO-*d*_6_) *δ* ppm 8.24 (d, *J =* 7.4 Hz, 2H, Ar-H), 8.01 (s, 1H, Ar-H), 7.82–7.70 (m, 10H, Ar-H), 7.62–7.60 (m, 3H, Ar-H), 7.43 (s, 1H, OH), 3.79 (s, 1H, NH), 3.19 (s, 3H, CH_3_). ^13^C NMR (101 MHz, DMSO-*d*_6_) *δ* ppm 163.18, 158.68, 150.89, 148.67, 144.03, 142.48, 135.60, 134.65, 134.41, 134.34, 133.53, 130.74, 129.79, 129.37, 128.80, 127.88, 127.67, 126.98, 126.91, 126.72, 119.16, 21.80. Anal. Calcd for C_28_H_22_N_4_O_2_ (446.50): C, 75.32; H, 4.97; N, 12.55. Found: C, 75.21; H, 5.08; N, 12.75.

##### 4-(6-((*Z*)-((*E*)-(4-Methoxybenzylidene)hydrazono)(phenyl)methyl)-1*H*-benzo[d]imidazol-2-yl)phenol (**3m**)

Yield: 0.36 g (82%); mp: 191–193 °C, ^1^H NMR (400 MHz, DMSO-*d*_6_) *δ* ppm 10.39 (s, 1H, OH), 8.64 (s, 1H, HC=N), 8.08 (d, *J =* 7.8 Hz, 2H, Ar-H), 7.96 (s, 1H, Ar-H), 7.83–7.58 (m, 7H, Ar-H), 7.06 (d, *J* = 7.8 Hz, 2H, Ar-H), 7.00 (d, 4H, Ar-H), 4.20 (s, 1H, NH), 3.83 (s, 3H, OCH_3_). ^13^C NMR (101 MHz, DMSO-*d*_6_) *δ* ppm 163.93, 161.43, 152.57, 148.09, 147.61, 136.24, 135.42, 133.17, 132.33, 131.90, 130.72, 129.58, 128.79, 128.21, 127.17, 126.83, 123.25, 115.69, 114.44, 114.15, 55.91. Anal. Calcd for C_28_H_22_N_4_O_2_ (446.50): C, 75.32; H, 4.97; N, 12.55. Found: C, 75.19; H, 4.73; N, 12.80.

##### 4-(5-((*Z*)-((*E*)-(1-(4-Methoxyphenyl)ethylidene)hydrazono)(phenyl)methyl)-1*H*-benzo[d]imidazol-2-yl)phenol (**3n**)

Yield: 0.36 g (79%); mp: 167–169 °C,^1^H NMR (400 MHz, DMSO-*d*_6_) *δ* ppm 10.60 (s, 1H, OH), 8.08 (d, *J =* 7.0 Hz, 3H, Ar-H), 8.00 (s, 1H, Ar-H), 7.85–7.73 (m, 7H, Ar-H), 7.60 (t, 3H, Ar-H), 7.05 (d, 3H, Ar-H), 4.33 (s, 1H, NH), 3.96 (s, 3H, OCH_3_), 2.51 (s, 3H, CH_3_).^13^C NMR (101 MHz, DMSO-*d*_6_) *δ* ppm 162.17, 160.94, 152.42, 153.40, 138.06, 137.40, 137.03, 132.93, 130.47, 130.03, 130.00, 129.52, 129.01, 126.99, 125.98, 118.00, 117.80, 115.20, 114.87, 55.86, 56.49, 21.59. Anal. Calcd for C_29_H_24_N_4_O_2_ (460.53): C, 75.63; H, 5.25; N, 12.17. Found: C, 75.61; H, 5.15; N, 12.25.

##### 4-(6-((*Z*)-((*E*)-(4-chlorobenzylidene)hydrazono)(phenyl)methyl)-1*H*-benzo[d]imidazol-2-yl)phenol (**3o**)

Yield: 0.35 g (78%); mp: 159–161 °C, ^1^H NMR (400 MHz, DMSO-*d*_6_) *δ* ppm 10.33 (s, 1H, OH), 8.07 (d, *J =* 7.8 Hz, 3H, HC=N and Ar-H), 7.95 (s, 1H, Ar-H), 7.90 (d, *J =* 7.0 Hz, 1H, Ar-H), 7.79–7.57 (m, 10H, Ar-H), 7.00 (d, *J* = 7.8 Hz, 3H, Ar-H), 4.20 (s, 1H, NH). ^13^C NMR (101 MHz, DMSO-*d*_6_) *δ* ppm 164.21, 156.38, 152.78, 148.09, 147.17, 146.59, 135.95, 135.60, 134.93, 133.80, 132.80, 129.99, 129.69, 129.42, 129.19, 129.09, 128.90, 128.81, 122.95, 117.89, 113.00. Anal. Calcd for C_27_H_19_ ClN_4_O (450.92): C, 71.92; H, 4.25; N, 12.43. Found: C, 71.99; H, 4.53; N, 12.49.

### 5.2. Biological Evaluation

#### 5.2.1. NCI Screening Assay

As mentioned, the methodology of the NCI procedure for the primary anticancer assay was detailed on their site (http://www.dtp.nci.nih.gov, accessed on 1 January 2020). However, briefly, the protocol was performed in full panels of 60 human tumor cell lines derived from 9 different neoplastic diseases. NCI-60 testing is performed in two parts: first, a single concentration is tested in all 60 cell lines at a single dose of 10^−5^ molar or 15 µg/mL in accordance with the protocol of the Drug Evaluation Branch, National Cancer Institute, 37 Convent Dr, Bethesda, MD 20814, USA. If the results obtained meet the selection criteria, then, second, the compound is tested again in all 60 cell lines in 5 × 10-fold dilution with the top dose being 10^−4^ molar or 150 µg/mL.

#### 5.2.2. MTT Assay for Cell Viability

To investigate the effect of the newly synthesized compounds on lung cancer cells, an MTT assay was performed against A549 and NCI-H460 cell lines (see Appendix B).

#### 5.2.3. VEGFR Inhibitory Assay

A VEGFR-2 assay was performed by the established reported method using a VEGFR-2 (KDR) Kinase Assay Kit (Catalog # 40325, BPS Bioscience, Biotechnology company, San Diego, CA, USA) for selected synthetic compounds **3e** and **3g**. Details are summarized in Appendix B.

#### 5.2.4. Cell Cycle Analysis and Apoptotic Assay

##### Cell Apoptosis and Apoptotic Detection

Studies on the effect of compound **3e** and **3g** on cell cycle development and induction of apoptosis in the NCI-H460 lung cell wetr done using the fluorescent Annexin V-FITC/ PI detection kit (BioVision EZCellTM Cell Cycle Analysis Kit Catalog #K920, Milpitas Blvd., Milpitas, CA 95035 USA) by flow cytometry assay. For more details, see Appendix B.

#### 5.2.5. Activation of Caspases

For more a deeper and more systematic investigation on cell apoptosis, the effect of compounds **3e** and **3g** on caspase-9 was evaluated compared to control. Details are summarized in Appendix B.

#### 5.2.6. Docking Study

Molecular Operating Environment (MOE) version 2021 is used to perform the molecular modeling study. The structure of **3g** was built in the MOE database (see Appendix B).

## Data Availability

The raw/processed data generated in this work are available upon request from the corresponding author.

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
