# Peer review of "Design, Synthesis, Docking Study, and Antiproliferative Evaluation of Novel Schiff Base–Benzimidazole Hybrids with VEGFR-2 Inhibitory Activity"

_molecules, 2023, doi:10.3390/molecules28020481_

Round 1

Reviewer 1 Report

The authors prepared a new series of Schiff-benzimidazole hybrids and characterize their structures by different spectroscopic and elemental analysis. Furthermore, the target compounds were evaluated for their in vitro cytotoxic activity against 60 cancer cell lines according to NCI single and five doses protocols. Huge work has been done. Interestingly, the binding interactions of these 35 compounds were confirmed through molecular docking studies. I find the paper very interesting to the readers of Molecules, and thus I am inclined to recommend publication, whereas, the following aspect should be promoted after minor revision.

1, Page 4-line 119, please check the formula of hydrazine hydrate. Page 3-line 105, please check the expression of “in good to excellent yield”. Page 7-line 239, the compound name of 3e and 3g was not in bold, including the compound name in the 5. Experimental part, please check all over the manuscript weather all the compound names are in bold.

2, the pictures of Figure 2, Figure 3A, 4A, 5 are ugly, the unit of uM in the labels under the Histogram are not clear ,and labels should be simplified to make the picture clear and beautiful. Moreover, the pictures of3D models of Figure 6 and Figure 7 are not clear, please redraw them.

3, please check the abbreviations of journal name in the ref1, ref4, ref20.

4, Figure S32-S45 are not clear.

Author Response

18 Dec 2022

To,

Molecules editorial office
MOLECULES

Ms. Ref. No.:  Molecules-2104860

Manuscript Title:

Design, Synthesis, Docking Study and Antiproliferative Evaluation of Novel Schiff Base-Benzimidazole Hybrids with VEGFR-2 Inhibitory Activity

Dear Editor,

We are thankful to you, the reviewers, and the editorial office for giving valuable time to our manuscript. We are thankful to the reviewers for suggesting very good suggestions and improving the quality of our manuscript. Referring to the above matter, below are our responses to the reviewers’ comments and suggestions.

Reviewers' comments:

Reviewer #1:

The authors prepared a new series of Schiff-benzimidazole hybrids and characterize their structures by different spectroscopic and elemental analyses. Furthermore, the target compounds were evaluated for their in vitro cytotoxic activity against 60 cancer cell lines according to NCI single and five doses protocols. Huge work has been done. Interestingly, the binding interactions of these 35 compounds were confirmed through molecular docking studies. I find the paper very interesting to the readers of Molecules, and thus I am inclined to recommend publication, whereas, the following aspect should be promoted after minor revision.

The authors highly appreciate the reviewer’s valuable and supportive comments that have emerged after his precise revision, which would help to improve the quality of this whole manuscript. Herein, we will reply to all points one by one as follows

  • Page 4-line 119, please check the formula of hydrazine hydrate. Page 3-line 105, please check the expression of “in good to excellent yield”. Page 7-line 239, the compound name of 3e and 3g was not in bold, including the compound name in the 5. Experimental part, please check all over the manuscript whether all the compound names are in bold.

Response: The authors welcome this suggestion from the reviewer and thank him for his deep revision and his concern.  Accordingly, the pointed sentence and typos were corrected as advised.

  • The pictures of Figure 2, Figure 3A, 4A, 5 are ugly, the unit of uM in the labels under the Histogram are not clear and labels should be simplified to make the picture clear and beautiful. Moreover, the pictures of3D models of Figure 6 and Figure 7 are not clear, please redraw them.

Response: The authors welcome this suggestion from the reviewer. Accordingly, all these pictures were changed to more clear ones.

  • Please check the abbreviations of the journal name in the ref1, ref4, ref20.

Response: The authors are very thankful to the respected reviewer, after a recheck of these journals’ abbreviations; they seem correct.

  • Figures S32-S45 are not clear

Response: The authors welcome this suggestion from the review; consequently, an updated supplementary file with more clear pictures was attached to the system.

All corrections highlighted in red color

We anticipate that our work on reviewers’ suggestions and comments will satisfy their queries and we hope your kind consideration on the revised manuscript to be published in your prestigious MOLECULES 

 Yours sincerely,

The corresponding author

Reviewer 2 Report

I suggest the following modifications to increase the quality of the manuscript: although the references in the introduction are adequate, the text is not clear regarding the intention of the research and does not clearly show the objective of the manuscript. Regarding the term Schiff base and especially azomethine, it should be reviewed since the functional group involves a second nitrogen atom, being more appropriate to use the term hydrazone. Figures 2 and 3 are not clear and it is suggested to improve them, specifically making the lines less thick and allowing to see the value of the SD in each bar. It is suggested to carry out a complementary study by molecular dynamics that helps to explain the results of molecular docking and certain biological activities.

Author Response

18 Dec 2022

To,

Molecules editorial office
MOLECULES

Ms. Ref. No.:  Molecules-2104860

Manuscript Title:

Design, Synthesis, Docking Study, and Antiproliferative Evaluation of Novel Schiff Base-Benzimidazole Hybrids with VEGFR-2 Inhibitory Activity

Dear Editor,

We are thankful to you, the reviewers, and the editorial office for giving valuable time to our manuscript. We are thankful to the reviewers for suggesting very good suggestions and improving the quality of our manuscript. Referring to the above matter, below are our responses to the reviewers’ comments and suggestions.

Reviewers' comments:

Reviewer #2:

The authors highly appreciate the reviewer’s valuable and supportive comments that have emerged after his precise revision, which would help to improve the quality of this whole manuscript. Herein, we will reply to all points one by one as follows

  • I suggest the following modifications to increase the quality of the manuscript: although the references in the introduction are adequate, the text is not clear regarding the intention of the research and does not clearly show the objective of the manuscript.

Response: The authors welcome this suggestion from the review; accordingly, clarifying statement was highlighted in the introduction part.

  • Regarding the term Schiff base and especially azomethine, it should be reviewed since the functional group involves a second nitrogen atom, being more appropriate to use the term hydrazone.

Response: The authors welcome this suggestion from the reviewer and thank him for his deep revision and his concern.  Accordingly, the pointed term was corrected all over the manuscript as advised.

  • Figures 2 and 3 are not clear and it is suggested to improve them, specifically by making the lines less thick and allowing us to see the value of the SD in each bar.

Response: The authors welcome this suggestion from the reviewer, all manuscript figures were changed to more clear ones as advised.

  • It is suggested to carry out a complementary study by molecular dynamics that help to explain the results of molecular docking and certain biological activities

Response: The authors welcome this suggestion from the respected reviewer and thank him for it, but for the limitation of time and sources the authors had been content with molecular docking. A molecular dynamics study will be in our future plan.

All corrections highlighted in yellow

We anticipate that our work on reviewers’ suggestions and comments will satisfy their queries and we hope your kind consideration on the revised manuscript to be published in your prestigious MOLECULES 

Yours sincerely,

The corresponding author